# miR-145-5p Targets Sp1 in Non-Small Cell Lung Cancer Cells and Links to BMI1 Induced Pemetrexed Resistance and Epithelial–Mesenchymal Transition

**DOI:** 10.3390/ijms232315352

**Published:** 2022-12-05

**Authors:** Wen-Wei Chang, Bing-Yen Wang, Shih-Hong Chen, Peng-Ju Chien, Gwo-Tarng Sheu, Ching-Hsiung Lin

**Affiliations:** 1Department of Biomedical Sciences, Chung Shan Medical University, No. 110, Sec. 1, Jianguo N. Rd., Taichung City 402306, Taiwan; 2Department of Medical Research, Chung Shan Medical University Hospital, No. 110, Sec. 1, Jianguo N. Rd., Taichung City 402306, Taiwan; 3Division of Thoracic Surgery, Department of Surgery, Changhua Christian Hospital, No. 135, Nanhsiao Street, Changhua 500209, Taiwan; 4Department of Post-Baccalaureate Medicine, College of Medicine, National Chung Hsing University, No. 145, Xingda Rd., South Dist., Taichung City 402202, Taiwan; 5Institute of Medicine, Chung Shan Medical University, No. 110, Sec. 1, Jianguo N. Rd., Taichung City 402306, Taiwan; 6Division of Chest Medicine, Department of Internal Medicine, Changhua Christian Hospital, No. 135, Nanhsiao Street, Changhua 500109, Taiwan; 7Institute of Genomics and Bioinformatics, National Chung Hsing University, No. 145, Xingda Rd., South Dist., Taichung 402202, Taiwan; 8Ph.D. Program in Translational Medicine, National Chung Hsing University, No. 145, Xingda Rd., South Dist., Taichung 402202, Taiwan; 9Department of Recreation and Holistic Wellness, MingDao University, No. 369, Wen-Hua Rd., Pitou Township, Changhua 523008, Taiwan

**Keywords:** non-small lung cancer, pemetrexed, drug resistance, Sp1, BMI1, microRNA, miR-145-5p, epithelial–mesenchymal transition

## Abstract

Pemetrexed is a folic acid inhibitor used as a second-line chemotherapeutic agent for the treatment of locally advanced or metastatic non-small cell lung cancer (NSCLC), which accounts for 85% of lung cancers. However, prolonged treatment with pemetrexed may cause cancer cells to develop resistance. In this study, we found increased expressions of BMI1 (B Lymphoma Mo-MLV insertion region 1 homolog) and Sp1 and a decreased expression of miR-145-5p was found in pemetrexed-resistant A400 cells than in A549 cells. Direct Sp1 targeting activity of miR-145-5p was demonstrated by a luciferase based Sp1 3′-UTR reporter. Changed expression of miR-145-5p in A400 or A549 cells by transfection of miR-145-5p mimic or inhibitor affected the sensitivity of the cells to pemetrexed. On the other hand, the overexpression of Sp1 in A549 cells caused the decreased sensitivity to pemetrexed, induced cell migratory capability, and epithelial–mesenchymal transition (EMT) related transcription factors such as Snail Family Transcriptional Repressor 1 and Zinc Finger E-Box Binding Homeobox 1. In addition, the overexpression of BMI1 in the A549 cells resulted in an increase in Sp1 and a decrease in miR-145-5p accompanied by the elevations of cell proliferation and EMT transcription factors, which could be reduced by the overexpression of miR-145-5p or by treatment with the Sp1 inhibitor of mithramycin A. In conclusion, the results of this study suggest that the downregulation of miR-145-5p by BMI1 overexpression could lead to the enhanced expression of Sp1 to induce the EMT process in pemetrexed-resistant NSCLC cells. These results suggest that increasing miR-145-5p expression by delivering RNA drugs may serve as a sensitizing agent for pemetrexed-resistant NSCLC patients.

## 1. Introduction

Lung cancer is a malignant lung tumor and is commonly classified as small cell lung cancer or non-small cell lung cancer (NSCLC) [1], with the latter accounting for approximately 85% of cases [2]. In recent years, every 10 µg/m^3^ increase in air pollution PM2.5 suspended particulate concentration has increased lung cancer mortality by 15–20% [3]. Common symptoms of lung cancer include cough, weight loss, shortness of breath, and chest pain [4]. In 2020, lung cancer was the leading cause of cancer deaths worldwide with an estimated 1.8 million deaths [5]. The current treatment options for lung cancer include surgery, chemotherapy, and radiation therapy. According to the investigation by our group, the 5-year overall survival (OS) rate of lung cancer in Taiwan was 25.0% using the data in the Taiwan Cancer Registry. For the patients receiving any kind of treatment, the OS was 27.1% [6], which indicates that it is urgent to develop new treatments for lung cancer. In NSCLC, epidermal growth factor receptor (EGFR) mutations account for 15–40% of cases of non-squamous type and EGFR inhibitors have been used clinically to treat lung cancer caused by EGFR abnormalities [7].

Pemetrexed (trade name Alimta), a folic acid antagonist, is primarily used as a second-line agent for the treatment of advanced or metastatic NSCLC, and is also approved to use as a first-line agent for the treatment of malignant mesothelioma and non-squamous type of NSCLC [8]. The main mechanism of action of pemetrexed is the inhibition of glycinamide ribonucleotide formyltransferase, dihydrofolate reductase, and thymidylate synthase (TS), which are involved in the metabolism of folic acid and the enzymes of purine and pyrimidine [9]. However, long-term treatment of pemetrexed in NSCLC may lead to resistance due to the high expression of TS [10], which is also considered as a biomarker for NSCLC [11]. However, it has been shown that there was no correlation between TS expression in NSCLC tissues and the development of pemetrexed resistance in patients [12], suggesting that the mechanism of pemetrexed resistance is not fully understood. We previously demonstrated that B lymphoma Mo-MLV insertion region 1 (BMI1) was increased in pemetrexed resistant NSCLC cells and the inhibition of BMI1 by a small molecule inhibitor suppressed the cancer stem cell activity and increased the pemetrexed sensitivity of the pemetrexed resistant cells [13]. However, the signaling networks of BMI1 upregulation in pemetrexed resistant NSCLC cells are not fully understood.

miR-145-5p is a microRNA (miRNA) of 22 nucleotides in length and is the guide strand of pre-miR-145 [14,15]. miR-145-5p is widely regarded as a tumor suppressor miRNA [16], and it has been shown that miR-145-5p is less expressed in lung cancer cells than in normal tissues [17]. Wang et al. found that miR-145-5p regulated the Sp1/NF-B signal transduction pathway to inhibit the metastatic and invasive ability of squamous cell carcinoma [18]. miR-145-5p was shown to inhibit the proliferation and metastasis of bladder cancer by regulating Transgelin-2 [19] and to inhibit the growth, invasion, and metastasis of colorectal cancer cells by targeting Rhomboid domain containing 1 [20]. In NSCLC, miR-145-5p has been shown to target the c-Jun N-terminal kinase (JNK) signaling pathway mediated by Mitogen-activated protein kinase kinase kinase 1 (MAP3K1), leading to the inhibition of epithelial–mesenchymal transition (EMT) [21]. Zheng et al. reported that circPVT1 functioned as a competing endogenous RNA to downregulate miR-145-5p in A549 with the resistance to cisplatin and pemetrexed [21]. They also found that the direct target of miR-145-5p in the double resistant A549 cells was ATP binding cassette subfamily c member 1 (ABCC1), a cell surface glycoprotein to transport chemotherapeutic compounds in cancer cells [21]. In this study, we would like to investigate the possible connections of miR-145-5p, Sp1, and BMI1-induced EMT in the pemetrexed resistance of NSCLC cells.

## 2. Results

### 2.1. The Expression Levels of miR-145-5p Change Pemetrexed Sensitivity of NSCLC Cells

We have previously established pemetrexed resistant NSCLC cells by long-term treatment of pemetrexed in A549 cells, named as A400 [9]. Here, we confirmed the pemetrexed resistant phenotype of A400 cells (Figure 1A) and further discovered that the miR-145-5p expression was significantly decreased in the A400 cells (Figure 1B). Next, we manipulated the intracellular miR-145-5p levels of A549 or A400 cells by transfection of the miR-145-5p inhibitor (Figure 1C) or miR-145-5p mimic (Figure 1D), respectively, followed by examining the pemetrexed sensitivity. The results revealed that the suppression of miR-145-5p in A549 cells caused the decreased sensitivity of pemetrexed (Figure 1E), while the overexpression of miR-145-5p enhanced the proliferation inhibition activity of pemetrexed in A400 cells (Figure 1F). These data demonstrate that the expression levels of miR-145-5p influence the pemetrexed sensitivity in NSCLC cells.

### 2.2. Sp1 Is a Direct Target of miR-145-5p in NSCLC Cells

We previously reported that the increased protein levels of BMI1 and TS could be observed in pemetrexed resistant A400 cells [13] and here we also confirmed them (Figure 2A). In addition, we further discovered the upregulation of Sp1 in A400 cells when compared to the original A549 cells (Figure 2A). Due to the discovery of Sp1 involved in the cancer suppressive effects of miR-145 [22], we next examined the Sp1 protein level after manipulating the intracellular level of miR-145-5p in the NSCLC cells. The results found that the increased Sp1 could be found in A549 after the inhibition of miR-145-5p by transfecting a specific inhibitor, while the decreased Sp1 protein was found in A400 cells when overexpressed in miR-145-5p by transfecting a specific mimic (Figure 2B). With the comparison of the sequences between Sp1 3′-UTR and mature miR-145-5p, we found the putative miR-145-5p binding site (Figure 2C) and cloned the Sp1 3′-UTR into a pMirTarget 3′-UTR assay vector. After transfecting the Sp1 3′-UTR reporter vector into the A549 and A400 cells, the luciferase activity was higher in A400 than those of thee A549 cells (Figure 2D). The transfection of the miR-145-5p inhibitor into the A549 cells increased the luciferase activity (Figure 2E), while the delivery of thee miR-145-5p mimic into A400 cells decreased the luciferase activity (Figure 2F). The changes in the luciferase activity were abolished when the binding site of miR-145-5p was deleted in the Sp1 3′-UTR reporter (Figure 2E,F). These data demonstrate that Sp1 is a direct target of miR-145-5p in the NSCLC cells.

### 2.3. Overexpression of Sp1 Decreases Pemetrexed Sensitivity and Induction EMT in NSCLC Cells

We next investigated the effects of Sp1 overexpression in A549 cells. With the transfection of the Sp1 cDNA vector and selected by hygromycin B, we established a Sp1 overexpressed subline called A549-Sp1-S2. We first compared the pemetrexed sensitivity of A549-Sp1-S2 to the parental cells and found the decreased pemetrexed sensitivity after Sp1 overexpression (Figure 3A). EMT can be induced by Sp1 in pancreatic ductal adenocarcinoma cells [23] and the EMT program is known to promote drug resistance in cancer cells [24]. We further found that the cell migration capability was increased in A549-Sp1-S2 cells compared to the parental A549 cells (Figure 3B) with the downregulation of E-cadherin and upregulation of ZEB1 and Snail1 (Figure 3C). The increased expression of TS was also observed in thee A549-Sp1-S2 cells (Figure 3C). These data indicate that increased Sp1 expression in NSCLC cells could decrease pemetrexed sensitivity and induce the EMT program.

### 2.4. miR-145-5p Suppresses BMI1 Induced Cell Proliferation and EMT in NSCLC Cells

We have previously reported that the upregulation of BMI1 promoted pemetrexed resistance in the NSCLC cells [13]. Here, we confirmed that A549-BS1, the BMI1 overexpressed A549 cells, displayed a resistance to pemetrexed (Figure 4A). Next, we observed a decreased expression of miR-145-5p in the A549-BS1 cells and the treatment of PTC-209, a small molecule inhibitor of BMI1, induced miR-145-5p expression in the pemetrexed resistant A400 cells (Figure 4B). Based on these observations, we hypothesized that forced miR-145-5p expression might suppress BMI1 induced malignant phenotypes. We first examined the expression of Sp1 in A549-tRFP, the vector control cells with the expression of the tRFP fluorescent protein, and A549-BS1 cells with or without miR-145-5p overexpression and found that the overexpression of BMI1 increased Sp1 expression and could be suppressed by the forced expression of miR-145-5p (Figure 4C). BMI1 overexpression in A549 cells increased cell proliferation and was reduced by the delivery of the miR-145-5p mimic (Figure 4D). We also found that the cell migration capability (Figure 4E) and the EMT program (Figure 4F) were increased with BMI1 overexpression, but these phenomena were suppressed by the transfection of the miR-145-5p mimic. These data suggest that miR-145-5p could suppress BMI1 induced malignant phenotypes in NSCLC cells.

### 2.5. BMI1 Induced Malignant Phenotypes in NSCLC Cells Can Be Reduced by Sp1 Inhibition

Since we demonstrated that Sp1 could be a direct target of miR-145-5p, we next investigated whether the suppression of Sp1 activity could also reduce BMI1-induced malignant features in the NSCLC cells. With the treatment of mithramycin A, a Sp1 selective inhibitor, the induced cell proliferation by BMI1 overexpression was significantly reduced (Figure 5A). We also found that the BMI1-induced cell migration capability (Figure 5B) and EMT program (Figure 5C) could be decreased by the treatment of mithramycin A in a dose-dependent manner. Finally, we analyzed the data of the Cancer Genome Atlas (TCGA) by The Encyclopedia of RNA Interactomes (ENCORI) webtool and found a decreased expression of miR-145-5p in tumor samples of lung adenocarcinoma (LUAD) and lung squamous cell carcinoma subjects (LUSC) (Figure 6A). The analysis of the LUAD and LUSC datasets revealed a negative correlation between Sp1 and miR-145-5p, however, a significant difference was found in LUSC, while LUAD showed a *p*-value of 0.058 (Figure 6B). Using the Gene Expression Profiling Integrative Analysis (GEPIA) webtool for the analysis, we observed a strong positive correlation between Sp1 and BMI1 mRNA expression in the NSCLC samples of TCGA (Figure 6C). Although there was no prognostic role of BMI1, Sp1, or ZEB1 in the overall survival of NSCLC subjects with a combination of the LUAD and LUSC datasets (Appendix A), the subjects with high expression of a 3-gene signature consisted of BMI1, Sp1, and ZEB1 displayed a significantly shorter overall survival in the NSCLC dataset of TCGA (Figure 6D). It suggests that the EMT factors induced by BMI1 or Sp1 in the NSCLC cells may contribute to the poor outcome of patients. All of the data suggest that the malignant features in NSCLC with a high BMI1 expression level can be reduced by Sp1 inhibition or miR-145-5p overexpression.

## 3. Discussion

There were some limitations in this study. First, we did not obtain the clinical samples with pemetrexed resistance to prove the links among BMI1, Sp1, and miR-145-5p, which were identified in the cell line data. Second, the enhancing effects of the miR-145-5p mimics or Sp1 inhibitors in the pemetrexed sensitivity of NSCLC tumors have not been demonstrated in vivo. One could also notice that the increasing fold of miR-145-5p in the mimic transfected A400 cells reached a thousand times, but the changes in the cell viability under pemetrexed treatment were only 30–40% (Figure 1F). Thomson et al. previously demonstrated that the increasing fold of mature miRNA after the transfection of miRNA mimics usually reached a thousand times level. However, these miRNA mimics displayed little complexing with the Argonaute protein, which is required to form an RNA-induced silencing complex (RISC) to suppress the translation of target mRNAs [26]. The efficiency of forming functional RISC after the transfection of miRNA mimics may affect the following biological effects, which may not be relevant to the level of mature miRNA detected by the RT-qPCR method.

It has also been suggested that high expression of TS may serve as an indicator of the susceptibility of lung cancer cells to develop aa resistance to pemetrexed drugs [27]. However, the clinical study by He et al. found no significant correlation between the degree of TS expression and the responsiveness of NSCLC patients to pemetrexed [12]. Chang et al. also found that in the analysis of NSCLC patients treated with pemetrexed as a third- or fourth-line therapeutic agent, the expression level of TS was not significantly correlated with the clinicopathological factors [28]. These data suggest that the mechanisms of pemetrexed resistance in NSCLC cells were not solely dependent on the over-expression of TS. In our study, a high expression of TS was also found in the pemetrexed-resistant A400 cells compared to the original A549 cells, while the BMI1 and Sp1 expression also showed an increase (Figure 2A). In addition to TS, EMT may also be an important pathway to cause drug resistance. Liang et al. found that blocking the EMT pathway could eliminate resistance to anti-folate drugs in lung cancer cells [29]. Taken together, the mechanism of resistance to pemetrexed in lung cancer cells may arise from either the high expression of TS or inducing cells toward the EMT program.

Some studies have found that in NSCLC, the level of BMI1 expression is closely related to patient survival and tumor size [30,31]. In contrast, the high expression of BMI1 increases the self-renewal capacity of cancer cells [32] and may promote EMT [33] and in cancer stem cells (CSCs), BMI1 is also highly expressed [34]. In addition, several studies have found that the inhibition of BMI1 expression can reduce cancer resistance and suppress cancer progression [35,36]. In A549 cells, the expression of miR-145-5p was suppressed when BMI1 was highly expressed (Figure 4B). It has been found that the expression of miR-145 was negatively correlated with high expression of BMI1 in CSCs derived from SPC-A1 lung cancer cells [37]. In colorectal cancer cells, epigallocatechin-3-gallate, a polyphenol phytochemical, was discovered to inhibit the expression of Notch, BMI1, and EZH2 with an upregulation of the expression of miR-145, miR-34a, and miR-200c, thus reducing tumor growth [38]. These studies showed a negative correlation between BMI1 and miR-145 expression, and our results were consistent with these previous studies. In addition, we increased miR-145-5p in A549-BS1 cells with BMI1 overexpression by delivering miR-145-5p mimic oligos and found that the proliferation and migratory capabilities of NSCLC cells were inhibited (Figure 4D,E), indicating that miR-145-5p could suppress the BMI1-induced proliferation and metastatic potentials of NSCLC cells, suggesting that miR-145-5p RNA drugs have the potential to be developed as anti-cancer drugs in lung cancer patients with high level of BMI1 expression.

In A549 cells, when BMI1 was highly expressed, the inhibition of Sp1 activity by mithramycin A reduced cell proliferation (Figure 5A) and cell migration (Figure 5B), suggesting that cell proliferation and EMT induced by BMI1 overexpression required Sp1 activity. Wang et al. showed that BMI1 transcription was regulated by Sp1 and c-Myc in nasopharyngeal carcinoma (NPC) [39]. Zhang et al. also found that the inhibition of Sp1 expression could block the expression of BMI1, c-Myc, Klf4, and Oct4 in NPC [40]. These studies suggest that the transcription of BMI1 may be regulated by Sp1. In our study, however, we found that when BMI1 was overexpressed in A549 cells, the expression of Sp1 increased simultaneously (Figure 4C), indicating that there was an additional regulatory mechanism between BMI1 and Sp1 in the NSCLC cells than those found in the NPC cells [39]. In our study, the forced expression of BMI1 in the A549 cells upregulated Sp1 expression (Figure 4C). The high expression of BMI1 also caused the downregulation of miR-145-5p (Figure 4B), and the transfection of the miR-145-5p mimic led to a decrease in Sp1 expression in pemetrexed-resistant A400 cells (Figure 2B). These results suggest that BMI1 may induce Sp1 expression in NSCLC cells by inhibiting miR-145-5p, which ultimately increases proliferation and the EMT program in NSCLC and promotes cellular resistance to pemetrexed. Although BMI1 was negatively correlated with miR-145-5p expression in NSCLC cells, how BMI1 downregulates miR-145-5p expression still needs to be investigated. Our previous report has demonstrated that ZEB1 protein expression was increased in pemetrexed resistant A400 cells or in A549 cells with BMI1 overexpression [13] and we showed that miR-145-5p expression was downregulated in BMI1 overexpressed NSCLC cells in this study (Figure 4B). In breast CSCs, Polytarchou et al. reported that ZEB1 could suppress miR-145 expression [41]. Therefore, we hypothesize that the inhibition of miR-145-5p expression by BMI1 in pemetrexed resistant A400 cells might be achieved by inducing ZEB1 expression, but this hypothesis needs to be confirmed in the future.

## 4. Materials and Methods

### 4.1. Cell Culture

A549 cells were purchased from the American Type Culture Collection (ATCC, Manassas, VA, USA). A400 cells were pemetrexed resistance cells derived from A549 cells that were established as the previous report [9]. All cells were maintained in Dulbecco’s modified Eagle medium (DMEM, Gibco^TM^, Waltham, MA, USA) with 10% fetal bovine serum (FBS, Hyclone Laboratories Inc., South Logan, UT, USA), 1 mM glutamine (Gibco), 1 mM sodium pyruvate (Gibco), 1× penicillin/streptomycin/Amphotericin B (Gibco), and 1× non-essential amino acids (Gibco) at 37 °C, 5% CO_2_ incubator.

### 4.2. Determination of Pemetrexed Sensitivity

The sensitivity of pemetrexed was determined by the 3-(4,5-ddimethylthiazol-2-yl)-2,5-diphenyltetrazolium bromide (MTT) assay (Sigma-Aldrich, Saint Louis, MO, USA). A total of 1 × 10^3^ cells/well/100 μL in culture medium containing pemetrexed (1.5, 0.75, 0.375, 0.1875, 0.09375 μM) were seeded into 96-well plates and incubated at a 37 °C incubator with 5% CO_2_. After incubation for 72 h, 10 μL of MTT in a concentration of 1 mg/mL in ddH_2_O was added into each well followed by incubation for 2 h at 37 °C. The blue formazan crystals were dissolved with dimethyl sulfoxide (DMSO) at room temperature for 10 min. Finally, the absorbance at 570 nm was detected by a microplate reader (SpectraMax M5, Molecular Devices, San Jose, CA, USA).

### 4.3. Cell Transfection

For the transfection of the miR-145-5p mimic or inhibitor, 2 × 10^5^ cells were seeded in six-well plates with the DMEM medium containing 2% FBS and the miR-145-5p inhibitor, miR-145-5p mimic, or negative controls were mixed with INTEFERIN^TM^ transfection reagent (Polyplus, New York, NY, USA), following the manufacturer’s protocol. The miRNA oligos were synthesized by GenePharma (GenePharma, Shanghai, China) as the sequences listed below:

miR-145-5p inhibitor: 5′-AGGGAUUCCUGGGAAAACUGGAC-3′;

miR-145-5p mimic sense: 5′-GUCCAGUUUUCCCAGGAAUCCCU-3′, antisense: 5′- GGAUUCCUGGGAAAACUGGACUU-3′;

inhibitor negative control: 5′-CAGUACUUUUGUGUAGUACAA-3′;

mimic negative control sense: 5′-UUCUCCGAACGUGUCACGUTT-3′, antisense: 5′-ACGUGACACGUUCGGAGAATT-3′.

For the overexpression of Sp1 in A549 cells, the Sp1 cDNA containing vector was purchased from Sino Biological, Inc. (Beijing, China. Cat No. HG12024-UT) and transfected with TransIT-X2^TM^ Dynamic Delivery System (Mirus Bio LLC, Madison, WI, USA), following the manufacturer’s protocol. The stable clone of Sp1 overexpressed A549 was further established by the selection of 400 μg/mL of hygromycin.

### 4.4. Reverse Transcription Quantitative PCR (RT-qPCR)

The total cellular RNA were extracted by TRIzol^TM^ Reagent (Invitrogen, Carlsbad, CA, USA), according to the manufacturer’s protocol. The qPCR detection of miR-145-5p was conducted by using the Bulge-Loop^TM^ miRNA qRT-PCR Kit (RiboBio Co. Ltd., Guangzhou, China) with the manufacturer’s protocol. The reverse transcription of miR-145-5p was conducted by the RevertAid First Strand cDNA Synthesis Kit (Thermo Fisher Scientific, Waltham, MA, USA) with a miR-145-5p specific RT primer followed by SYBR Green based qPCR detection (SYBR^TM^ Green qPCR Supermixed, Bio-Rad, Hercules, CA, USA) with a miR-145-*p* specific forward primer and a universal reverse primer in a Eco 48 Real-Time PCR System (PCR max, Staffordshire, UK) with the following steps: 95 °C for 3 min, followed by 40 cycles of 95 °C for 10 s, 60 °C for 20 s, and 70 °C for 10 s. A melting analysis was performed to assess the specificity of PCR products.

### 4.5. Western Blotting

Cells were lyzed with the NETN buffer (20 mM Tris-HCl pH = 8.0, 150 mM NaCl, 1 mM EDTA, 0.5% NP-40). The total protein concentrations were quantified by the Bicinchoninic Acid Assay (BCA) Kit (Thermo Fisher Scientific). A total of 20 μg of total cellular proteins were loaded in 10% sodium dodecyl sulfate polyacrylamide gel for separation followed by transferring onto a polyvinylidene fluoride membrane (Pall Corporation, Washington, NY, USA). After blocking the membrane with 1% skim milk at room temperature for 1 h, the primary antibodies were then added and incubated at 4 °C overnight. On the next day, the membranes were washed with TBS-T (20 mM Tris pH = 8.0, 137 mM NaCl, 0.1% Tween-20), and added horseradish peroxidase conjugated secondary antibodies followed by incubated at room temperature for 1 h. The signals were then developed by Pierce^TM^ ECL Western Blotting Substrate (Thermo Fisher) and captured by Amersham^TM^ Imager 680 (Cytiva, Marlborough, MA, USA). The antibodies used in this study are provided in Appendix A.

### 4.6. Luciferase Reporter Assay

The Sp1 3′-UTR sequence that contained a predicted miR-145-5p binding site was constructed into a pMirTarget 3′-UTR assay vector (Origene Technologies, Inc., Rockville, MD, USA) after digesting with the restrict enzymes of EcoRI and XbaI. The mutant Sp1 3,’-UTR vector was constructed by deleting the predicted miR-145-5p binding site with a site-direct mutagenesis method using PCR synthesis of the wild type vector with mutant introducing primers followed by the digestion of the DpnI enzyme to remove the wild type sequences. The primer sequences for the constructions are provided in Appendix A. Co-transfection of the miR-145-5p mimic or inhibitor with the wild type or mutant luciferase reporter constructs was performed by the TransIT^TM^ X2 transfection reagent. After being incubated for 48 h, the firefly luciferase activities were measured by the Luciferase Reporter Assay System (Promega Corporation, Madison, WI, USA) and the firefly luciferase activity data were normalized by RFP signals after analyzing with a flow cytometer (BD FACSCalibur™ System, BD Biosciences, Franklin Lakes, NJ, USA.

### 4.7. Wound-Healing Assay

The migration ability of the cancer cells was measured by a wound-healing assay that was created by Ibidi GmbH (Grafelfing, Germany). Briefly, two silicone inserts were set up into each well of a 6-well plate followed by seeding 2 × 10^4^ cells in each side of the silicone insert for attachment. After incubating overnight, the silicone inserts were removed, washed with PBS twice, and replaced with fresh culture medium. The pictures of the wells were captured by SPOT Basic Software (version SOPT56B, SPOT Imaging, Sterling Heights, MI, USA) at 0, 3, 6, and 24 h post insert removal and the cell migration areas were analyzed by ImageJ software (Version 1.53a, National Institutes of Health, Bethesda, MD, USA).

### 4.8. Clonogenic Assay

The cell proliferation ability was evaluated by a clonogenic assay. A total of 2.5 × 10^2^ cells/well was seeded into 12-well plates and incubated for 5 days followed by fixation with 3.7% formaldehyde solution at room temperature for 10 min. After discarding the fixation solution, wells were washed with ddH_2_O twice, followed by staining with 0.5% crystal violet at room temperature for 2 h. The cell colonies were visualized as blue-purple and were pictured and counted.

### 4.9. Analysis of TCGA Data

The LUAD and LUSC datasets of the TCGA database were analyzed by two webtools including ENCORI [24] (accessed on 1 December 2022) and GEPIA2 [25] (accessed on 1 December 2022). In ENCORI, the number of cancer and normal in LUAD were 512 and 20, respectively. The cancer and normal numbers in the LUSC of ENCORI were 475 and 38, respectively. In GEPIA2, the number of cancer and normal in LUAD were 483 and 59, respectively. The cancer and normal numbers in LUSC in GEPIA2 were 486 and 50, respectively.

### 4.10. Statistical Analysis

Quantitative data were presented as the mean ±SD. The comparisons between two groups were analyzed with the Student’s *t*-test. The comparisons among multiple groups (more than two) were analyzed with one-way ANOVA followed by Tukey–Kramer’s post hoc test to identify differences in each two groups among all compared groups using Prism (version 5.0, GraphPad Software Inc., San Diego, CA, USA). A *p* value less than 0.05 was considered as statistically significant.

## 5. Conclusions

In conclusion, we found that miR-145-5p is involved in the resistance mechanism of NSCLC cells to pemetrexed treatment through the inhibition of Sp1, while miR-145-5p is negatively regulated by BMI1 in the NSCLC cells. The results of this study suggest that the expression level of miR-145-5p may be an indicator of the resistance to pemetrexed in NSCLC, and RNA drugs that enhance the miR-145-5p level in NSCLC tissues might have the potential to enhance the efficacy of pemetrexed, especially in subjects with a high level of BMI1 expression.

## Figures and Tables

**Figure 1 ijms-23-15352-f001:**
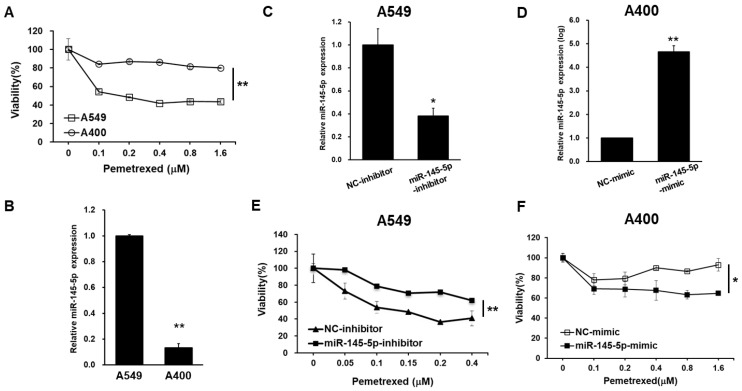
The miR-145-5p expression level in NSCLC cells affects the pemetrexed sensitivity. (**A**) The differential pemetrexed sensitivity between A549 cells and the pemetrexed resistant A400 cells was determined by the MTT assay. (**B**) The miR-145-5p expression levels in the A549 or A400 cells were measured by the RT-qPCR method. (**C**,**E**) The miR-145-5p inhibitor or negative control (NC) inhibitor was transfected in the A549 cells. The inhibition of miR-145-5p was confirmed by the RT-qPCR method (**C**) and the effect on pemetrexed sensitivity was determined by the MTT assay (**E**). (**D**,**F**) Overexpression of miR-145-5p in the A400 cells was conducted by the transfection of the miR-145-5p mimic and was confirmed by thee RT-qPCR method (**D**). Pemetrexed sensitivities of the miR-145-5p mimic or negative control (NC) mimic transfected A400 cells were further measured by the MTT assay (**F**). * *p* < 0.05; ** *p* < 0.01.

**Figure 2 ijms-23-15352-f002:**
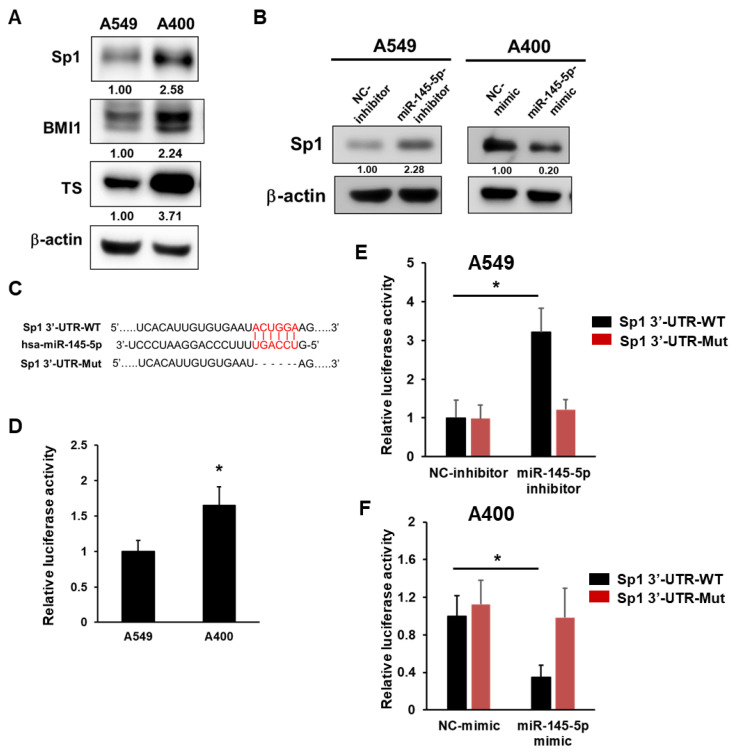
Sp1 is a direct target of miR-145-5p in the NSCLC cells. (**A**) The expression levels of Sp1, BMI1, or TS were determined by Western blotting. (**B**) The A549 or A400 cells were transfected with the miR-145-5p inhibitor or miR-145-5p mimic, respectively, and the expression level of Sp1 was determined by Western blotting. NC, negative control. (**C**) Alignments of miR-145-5p to wild type (Sp1 3′-UTR-WT) or mutant (Sp1 3′-UTR-Mut) of Sp1 3′-UTR. (**D**) A luciferase reporter vector containing Sp1 3′-UTR-WT sequences was transfected into A549 or A400 cells. The firefly luciferase activity was measured at 48 h post transfection and data were presented as the relative luciferase activity of A549 after being normalized with the RFP positive percentage. (**E**) A549 (**E**) or A400 (**F**) cells were transfected with the miR-145-5p inhibitor or miR-145-5p mimic, respectively, together with the Sp1 3′-UTR reporter vector simultaneously for 48 h. Luciferase activities were measured and presented as the relative level to negative control (NC) inhibitor/mimic group after normalization with the RFP positive percentage. * *p* < 0.05.

**Figure 3 ijms-23-15352-f003:**
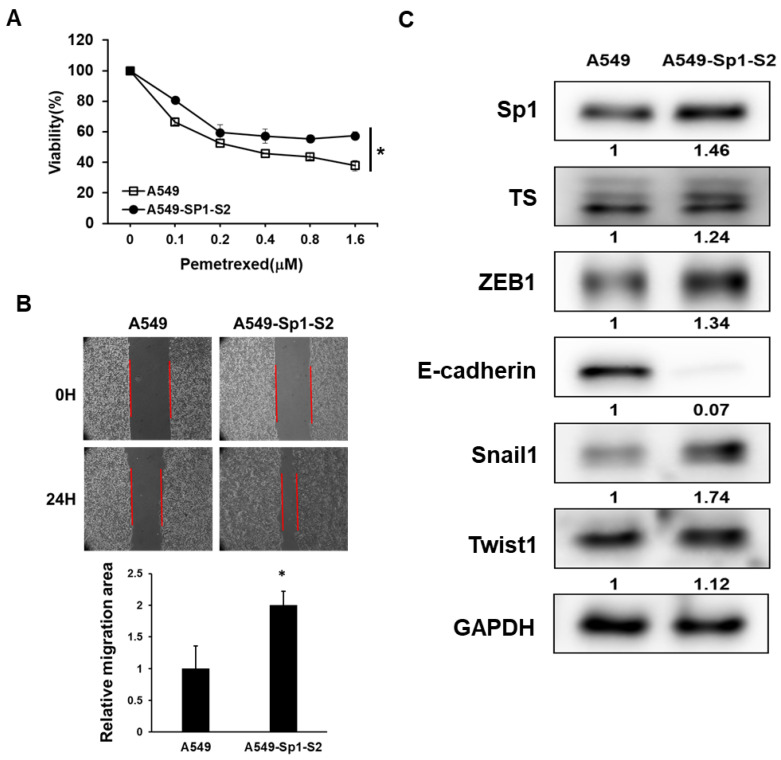
Forced expression of Sp1 in A549 cells decreases pemetrexed sensitivity and induces EMT. A549 cells were transfected with Sp1 cDNA containing the vector and selected with 400 μg/mL hygromycin B to establish the Sp1 overexpressed subline (A549-Sp1-S2). (**A**) Pemetrexed sensitivity was measured by the MTT assay. (**B**) Cell migration capability was determined by the wound healing assay and the cell migration area was measured by ImageJ software (version 1.53u, National Institutes of Health, Bethesda, MD, USA) at 24 h after the removal of the inserts. Red lines indicated the forefront of cell migration. Data are presented as the relative migration area to the A549 cells. (**C**) The protein expressions of Sp1, TS, ZEB1, E-cadherin, Snail1, and Twist1 were determined by Western blotting. The inserted numbers indicated relative expression level to A549 cells. * *p* < 0.05.

**Figure 4 ijms-23-15352-f004:**
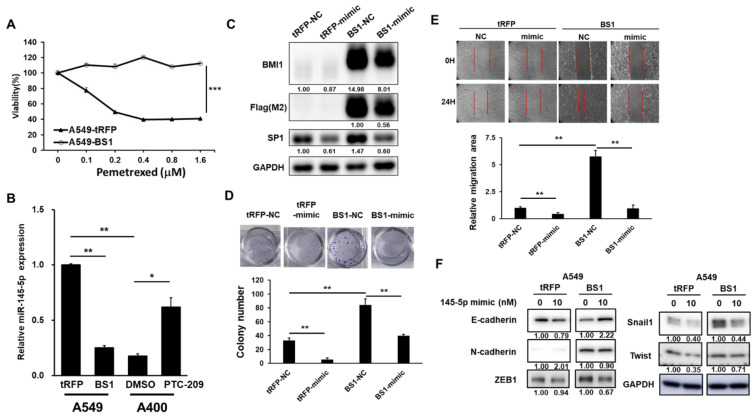
miR-145-5p suppresses BMI1-induced malignant phenotypes in A549 cells. BMI1 overexpressed A549 cells (A549-BS1) or tRFP expressing A549 cells (A549-tRFP) were established as per our previous report [13]. (**A**) Pemetrexed sensitivity was measured by the MTT assay. (**B**) Expressions of miR-145-5p in the A549 or A549-BS1 cells were determined by the RT-qPCR method. A400 cells were treated with 5 μM of PTC-209 for 48 h followed by the detection of miR-145-5p by RT-qPCR. (**C**) The expressions of BMI1, Sp1, or flag-tagged proteins were determined by Western blotting. (**D**,**E**) A549 or A549-BS1 cells were transfected with the miR-145-5p mimic or negative control (NC) mimic for 24 h and harvested cells to determine the cell proliferation capability by the clonogenic assay (**D**) or cell migration activity by the wound healing assay (**E**). Red lines indicated the forefront of cell migration. (**F**) A549-tRFP or A549-BS1 cells were transfected with the 10 nM miR-145-5p mimic or negative control mimic (NC, indicated as 0) for 48 h. The protein expressions of E-cadherin, N-cadherin, Snail1, or ZEB1 were determined by Western blotting. The inserted numbers indicate the relative expression levels compared to the NC mimic transfected cells. * *p* < 0.05; ** *p* < 0.01; *** *p* < 0.001.

**Figure 5 ijms-23-15352-f005:**
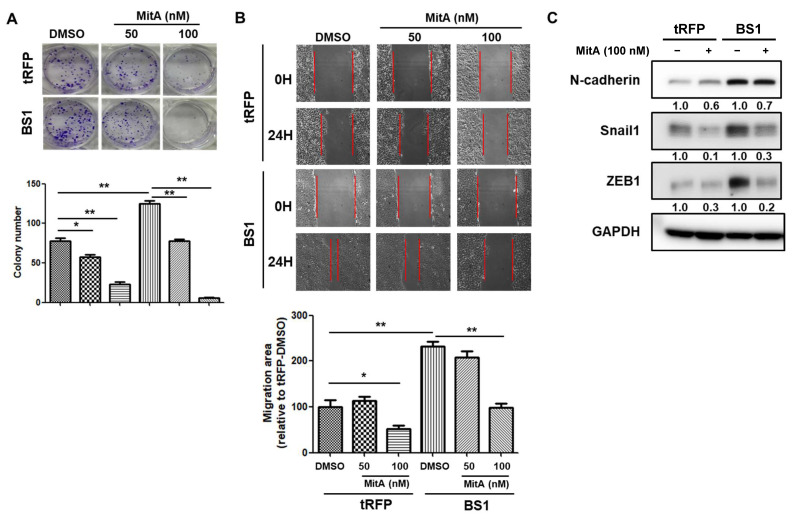
The inhibition of Sp1 activity reduces BMI1-induced malignant phenotypes in A549 cells. A549 or the derived BMI1 overexpressed A549-BS1 cells were treated with mithramycin A (MitA) at the concentrations of 50 or 100 nM. (**A**) Cell proliferation capability was measured by the clonogenic assay for 14 days. Cell colonies were visualized by the crystal violet stain. The statistical analysis was performed by one-way ANOVA with Tukey–Kramer’s post hoc test. * *p* < 0.05; ** *p* < 0.01. (**B**) Cell migration activity was determined by the wound healing assay and the cell migration areas were measured by ImageJ software. Red lines indicated the forefront of cell migration. * *p* < 0.05; ** *p* < 0.01. The statistical analysis was conducted as described in (**A**). (**C**) The expressions of N-cadherin, Snail1, or ZEB1 in A549-tRFP or A549-BS1 cells after treatment with 100 nM of MitA were determined by Western blotting. Inserted numbers indicated the relative expression level compared to the non-MitA treated groups after quantifying the band intensities by ImageJ software and using the non-treated sample as a reference.

**Figure 6 ijms-23-15352-f006:**
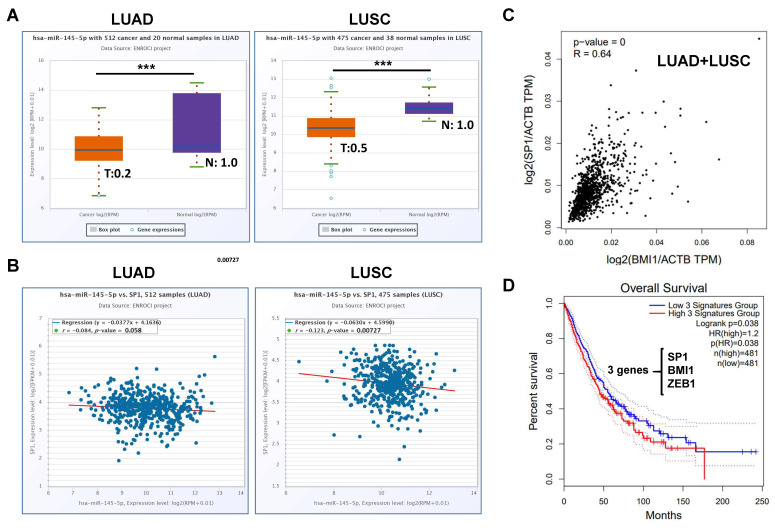
The correlations of BMI1, Sp1, and miR-145-5p in the lung cancer data of TCGA. (**A**) The RNA expression levels of miR-145-5p in normal lung tissues or cancer tissues of lung adenocarcinoma (LUAD) or lung squamous cell carcinoma (LUSC) were obtained from the TCGA database and analyzed using the ENCORI webtool [24]. T, tumor; N, normal. The number indicates the relative fold change in the tumor samples using normal samples as the reference. The statistical analysis was performed by the unpaired *t*-test. ***, *p* < 0.001. (**B**) The correlations between miR-145-5p and Sp1 mRNA expression in LUAD or LUSC were obtained from ENCORI and statistical analysis was performed by Spearman’s correlation. (**C**) The correlation between BMI1 and Sp1 mRNA in the NSCLC subjects (LUAD + LUSC) in the TCGA database was obtained from the GEPIA2 website [25] and the statistical analysis was performed by Spearman’s correlation. (**D**) Survival analysis based on a 3-gene-signature consisted of SP1/BMI1/ZEB1 in the NSCLC subjects of the TCGA database was analyzed using the GEPIA2 webtool with a cutoff set by the median expression level. The statistical analysis was performed by a log-rank test.

## Data Availability

Not applicable.

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
