# Peer review of "miR-145-5p Targets Sp1 in Non-Small Cell Lung Cancer Cells and Links to BMI1 Induced Pemetrexed Resistance and Epithelial–Mesenchymal Transition"

_ijms, 2022, doi:10.3390/ijms232315352_

Round 1

Reviewer 1 Report

Pg 2, line 62 - missing word:  "which indicates the urgent ____ for improving treatments in lung cancer"

Pg 2, line 84 - Please provide a few references for the following statement: "miR-145-5p is widely regarded as a tumor suppressor 84 miRNA"

Pg 2, line 89 - the "or" should be "and"

Pg 3, line 128 - this line does not belong in the results: "In gastric cancer, overexpression of miR-145 inhibited 128 cell proliferation, invasiveness, and cell cycle progression by inhibition of Sp1"

Figures 2-5: Could include size markers for western blots

Page 5, line 160-161 - These are not cited correctly (the formatting): EMT can be induced by Sp1 in pancreatic ductal adenocar- 160 cinoma cells (PMID: 30976063) and EMT programme is known to promote drug resistance 161 in cancer cells (PMID: 28397828).

Figure 6: Could you show or at least comment on the differences in survival via Kaplan Meier curves for the 4 genes individually and not just as an overall signature. 

Page 8, lines 246-7 - hese are not cited correctly (the formatting). It has also been suggested that high expression of TS may serve as an indicator of the 246 susceptibility of lung cancer cells to develop resistance to pemetrexed drugs (PMID: 247 19811498, PMID: 21487406).

Discussion: Good summary of salient features of the article. However, please reflect on and discuss briefly the weaknesses of your study. 

Author Response

Pg 2, line 62 - missing word:  "which indicates the urgent ____ for improving treatments in lung cancer"

Responses:

We thank the comments from the reviewer. The sentence has been rewritten as “which indicates it is urgent to develop new treatments for lung cancer.”.

Pg 2, line 84 - Please provide a few references for the following statement: "miR-145-5p is widely regarded as a tumor suppressor 84 miRNA"

Responses:

We add a new reference according to the suggestion.

Pg 2, line 89 - the "or" should be "and"

Responses:

It has been revised according to the suggestion.

Pg 3, line 128 - this line does not belong in the results: "In gastric cancer, overexpression of miR-145 inhibited 128 cell proliferation, invasiveness, and cell cycle progression by inhibition of Sp1"

Responses:

We revised the descriptions as follows: “Due to the discovery of Sp1 involving in the cancer suppressive effects of miR-145 [24], we next examined the Sp1 protein level after manipulating the intracellular level of miR-145-5p in NSCLC cells.”.

Figures 2-5: Could include size markers for western blots

Responses:

The size markers of all the western blot data have been included in the supporting information, which is a requirement for submission.

Page 5, line 160-161 - These are not cited correctly (the formatting): EMT can be induced by Sp1 in pancreatic ductal adenocar- 160 cinoma cells (PMID: 30976063) and EMT programme is known to promote drug resistance 161 in cancer cells (PMID: 28397828).

Responses:

We apologize for the unformatted citations and they have been corrected in this revisions.

Figure 6: Could you show or at least comment on the differences in survival via Kaplan Meier curves for the 4 genes individually and not just as an overall signature.

Responses:

The survival analyses using single gene of BMI1, Sp1, or ZEB1 did not show prognostic value but combination of these three genes achieved a significantly prognostic role. It suggests that the EMT induced by BMI1 or Sp1 in NSCLC cells may contribute to the poor outcome of patients. The data of survival analyses by single gene were included as a Figure S1. The Figure 6D was revised as 3-gene signature.

Page 8, lines 246-7 - hese are not cited correctly (the formatting). It has also been suggested that high expression of TS may serve as an indicator of the 246 susceptibility of lung cancer cells to develop resistance to pemetrexed drugs (PMID: 247 19811498, PMID: 21487406).

Responses:

We apologize for the unformatted citations and they have been corrected in this revisions.

Discussion: Good summary of salient features of the article. However, please reflect on and discuss briefly the weaknesses of your study.

Responses:

We thank the suggestion from reviewer that give us a chance to think about the weaknesses of our study. The descriptions of weaknesses have been included in the first paragraph of Discussion section as follows: “There are some limitations of this study. First, we did not obtain the clinical samples with pemetrexed resistance to prove the links among BMI1, Sp1, and miR-145-5p which were identified in cell line data. Second, the enhancing effects of miR-145-5p mimics or Sp1 inhibitors in pemetrexed sensitivity of NSCLC tumors has not been demonstrated in vivo. One may also notice that the increasing fold of miR-145-5p in mimic transfected A400 cells reached to thousand times, but the changes in cell viability under pemetrexed treatment were only 30-40% (Figure 1F). Thomson et al. previously demonstrated that the increasing fold of mature miRNA after transfection of miRNA mimics usually reached a thousand times level. However, these miRNA mimics displayed little complexing with Argonaute protein, which is required to form an RNA-induced silencing complex (RISC) to suppress the translation of target mRNAs [26]. The efficiency of forming functional RISC after transfection of miRNA mimics may affect the following biological effects, which may not be relevant to the level of mature miRNA detected by RT-qPCR method.”.

Reviewer 2 Report

The nanuscript titled “miR-145-5p targets Sp1 in non-small cell lung cancer cells and links to BMI1 induced pemetrexed resistance and epithelial-mesenchymal transition” authored by Wen-Wei Chang et al. investigated the signaling network leading to the acquisition of pemetrexed resistance in NSCLC and found connections between miR145-5p, Sp1,BMPI1, and EMT induction during development of pemetrexed resistance in NSCLC cells using forced expression or silencing by inhibitors of these signaling molecules in cell culture. They also found these correlations in human lung cancer using in silico analysis. The authors reported that increasing the expression of miR-145-5p by RNA drug delivery leads to suppression of Sp1 and BMI1 may act as a potent sensitizing agent in patients with pemetrexed-resistant NSCLC.

Overall, the objectives of the study were clear, and the experimental design was well-organized. Moreover, the conclusions based on their data were reasonable. There are a few minor collections and questions that I would like to ask the authors, described as follows;

(1) In the Results section (lines 106-111), the panel order in Figure 1 does not match the text’s description. I would suggest switching panel marks D and E in Figure 1.

(2) In Figure 1, the vertical axis of E is a logarithmic scale while C is not, so the expression difference is approximately 10,000 times compared to the control in E, but the difference in the viability on D and F is significant but not much, despite the large difference in expression levels; some description of this phenomenon may be required.

(3) For analyses using web tools based on public databases showing the correlation of BMI1, Sp1, and miR145-5p in Figure 6, the number of data points and the method of statistical analysis should also be described in the Materials and Methods section or include a Reference. Explanations of the category names of T, N, and the meaning of their numbers are missing in Figure 5A. Figure 5C also lacks a description of the methods used for statistical analysis.

(4) In the left panel in Figure 6B shows the relationship between SP1 and miR145-5p expression in LUAD data, with a p-value of 0.058. This does not meet the criterion that the author mentioned in the Methods section that p<0.05 is considered significant, but the text says that “there was a significant negative correlation between Sp1 and miR-145-5p” in lines 217-218, authors need to explain on this point.

(5) The resolution of the raster images in Figure 6A and B was low, and it was difficult to read the text in the figure. Please increase the resolution or use line-vector images.

(6) Other minor points
lines 161 and 162, the reference should be indicated in the reference number, not the PMID.

line 179, and the reference number is a blank. please check it.

line 185, a description of tRFP-A459 is missing. Please indicate what it is.

Author Response

Reviewer2

The manuscript titled “miR-145-5p targets Sp1 in non-small cell lung cancer cells and links to BMI1 induced pemetrexed resistance and epithelial-mesenchymal transition” authored by Wen-Wei Chang et al. investigated the signaling network leading to the acquisition of pemetrexed resistance in NSCLC and found connections between miR145-5p, Sp1,BMPI1, and EMT induction during development of pemetrexed resistance in NSCLC cells using forced expression or silencing by inhibitors of these signaling molecules in cell culture. They also found these correlations in human lung cancer using in silico analysis. The authors reported that increasing the expression of miR-145-5p by RNA drug delivery leads to suppression of Sp1 and BMI1 may act as a potent sensitizing agent in patients with pemetrexed-resistant NSCLC.

Overall, the objectives of the study were clear, and the experimental design was well-organized. Moreover, the conclusions based on their data were reasonable. There are a few minor collections and questions that I would like to ask the authors, described as follows;

Responses:

We first thank the positive feedbacks from the reviewers. The point-by-point revisions are listed below.

(1) In the Results section (lines 106-111), the panel order in Figure 1 does not match the text’s description. I would suggest switching panel marks D and E in Figure 1.
Responses:

We thank the comments from the reviewer. The marks of D and E in Figure 1 have been switched according to the suggestions.

(2) In Figure 1, the vertical axis of E is a logarithmic scale while C is not, so the expression difference is approximately 10,000 times compared to the control in E, but the difference in the viability on D and F is significant but not much, despite the large difference in expression levels; some description of this phenomenon may be required.
Responses:

It is usually to observe the thousand times increasing fold of mature miRNA after transfection of miRNA mimics, however, they may display little complexing with Argonaute protein (PloS one, 8(1), e55214. https://doi.org/10.1371/journal.pone.0055214). In other word, little miRNA mimics which being transfected into cells could function well. We think that is the reason for not huge difference in the changes of viablility despite the thousand times increasing in miR-145-5p level after transfection of mimics. We add some discussions about these phenomena in first paragraph of Discussion Section as follows: “One may notice that the increasing fold of miR-145-5p in mimic transfected A400 cells reached to thousand times, but the changes in cell viability under pemetrexed treat-ment were only 30-40% (Figure 1F). Thomson et al. previously demonstrated that the in-creasing fold of mature miRNA after transfection of miRNA mimics usually reached a thousand times level. However, these miRNA mimics displayed little complexing with Argonaute protein, which is required to form an RNA-induced silencing complex (RISC) to suppress the translation of target mRNAs [26]. The efficiency of forming functional RISC after transfection of miRNA mimics may affect the following biological effects, which may not be relevant to the level of mature miRNA detected by RT-qPCR method.”.

(3) For analyses using web tools based on public databases showing the correlation of BMI1, Sp1, and miR145-5p in Figure 6, the number of data points and the method of statistical analysis should also be described in the Materials and Methods section or include a Reference. Explanations of the category names of T, N, and the meaning of their numbers are missing in Figure 5A. Figure 5C also lacks a description of the methods used for statistical analysis.
Responses:

We apologize for the missing information of Figure 5 and Figure 6. They have been included in this revision.

(4) In the left panel in Figure 6B shows the relationship between SP1 and miR145-5p expression in LUAD data, with a p-value of 0.058. This does not meet the criterion that the author mentioned in the Methods section that p<0.05 is considered significant, but the text says that “there was a significant negative correlation between Sp1 and miR-145-5p” in lines 217-218, authors need to explain on this point.
Responses:

We revise the descriptions as follows: “Analysis of LUAD and LUSC datasets revealed a negative correlation between Sp1 and miR-145-5p, however, a significant difference was found only in LUSC, while LUAD showed a p-value of 0.058.”.

(5) The resolution of the raster images in Figure 6A and B was low, and it was difficult to read the text in the figure. Please increase the resolution or use line-vector images.
Responses:

We apologize for the low resolutions of Figure 6A and 6B and we have do our best to change they into high resolution ones.

(6) Other minor points
lines 161 and 162, the reference should be indicated in the reference number, not the PMID.
line 179, and the reference number is a blank. please check it.

Responses:

We apologize for the missing citations and they have been added in this revision.

line 185, a description of tRFP-A459 is missing. Please indicate what it is.

Responses:

We add the descriptions as follows: “…..A549-tRFP, the vector control cells with the expression of tRFP fluorescent protein, and ……”